# The Use of an Integrated Digital Tool to Improve the Efficiency of Multidisciplinary Tumor Boards—A Prospective Trial in Taiwan

**DOI:** 10.3390/cancers17030444

**Published:** 2025-01-28

**Authors:** Linda Chia-Fang Chang, Hsuan-Chih Kuo, Hung-Ming Wang, Yung-Chia Kuo, Ching-Ting Wang, Li-Chin Chen, Jason Chia-Hsun Hsieh

**Affiliations:** 1Cancer Center, New Taipei Municipal Tucheng Hospital, New Taipei City 236, Taiwan; lindacf@cgmh.org.tw (L.C.-F.C.); hsuanchihkuo@gmail.com (H.-C.K.); 2Department of Nursing, New Taipei Municipal Tucheng Hospital, New Taipei City 236, Taiwan; judy5612@cgmh.org.tw; 3Division of Hematology-Oncology, Department of Internal Medicine, New Taipei Municipal Tucheng Hospital, New Taipei City 236, Taiwan; 8705024@cgmh.org.tw; 4Department and College of Medicine, Chang Gung University, Taoyuan 333, Taiwan; whm526@cgmh.org.tw; 5Division of Hematology-Oncology, Department of Internal Medicine, Chang Gung Memorial Hospital at Linkou, Taoyuan 333, Taiwan; 6Department of Nursing, Chang Gung Memorial Hospital at Linkou, Taoyuan 333, Taiwan; gssunny@cgmh.org.tw

**Keywords:** navify tumor board, digital tool, multidisciplinary team tumor board, efficiency

## Abstract

Efficient decision-making in multidisciplinary tumor boards (MDTs) is critical for providing optimal cancer care, but traditional workflows can be time consuming and resource intensive. This study evaluates the navify Tumor Board, a digital tool designed to streamline MDT processes by integrating patient data and supporting collaborative decision-making. Our research aimed to assess whether this tool could significantly improve the efficiency of MDT workflows compared to conventional methods. By reducing time spent in preparation, discussion, and follow-up phases, the navify Tumor Board has the potential to enhance the productivity of cancer care teams, allowing them to focus more on patient outcomes. The findings from this study provide valuable insights for hospitals and medical professionals considering the adoption of digital solutions to optimize team-based cancer care, offering a framework for future research on innovative tools that can improve clinical efficiency and support better patient management.

## 1. Introduction

A multidisciplinary approach is considered one of the best ways to deliver complex care to patients with cancer [1]. Multidisciplinary care teams hold multidisciplinary tumor board (MDT) meetings regularly, in which several specialists collaborate to review individual patient data and make decisions regarding clinical care following the discussion. The benefits of tumor boards include an improvement in diagnosis and staging, changes to initial treatment plans, enhanced delivery of multimodal treatments, and integrated care from MDT members [2,3]. However, the workflow and preparation of MDTs are very cumbersome and time consuming. The difficulties are that information often resides in separate hospital systems such as electronic medical records (EMRs), laboratory systems, and imaging systems, which have to be collected and conformed to a format that is suitable for MDTs. This can lead to potential miscommunication, information being overlooked or duplicated, or the most updated information not being used [4,5]. Moreover, conducting an MDT meeting can also be challenged by time constraints, the inability of all members to attend, and increased administrative work [6,7]. Inefficiencies in the current MDT processes often increase the burden on the healthcare team, and extend the time taken to determine the best treatment plan for a patient [8].

The American Society of Clinical Oncology (ASCO) surveyed international ASCO members for suggestions to improve efficiency of MDTs. One of the most important suggestions was an improvement in infrastructure such as digital technologies to facilitate presentation and documentation [9]. The use and effectiveness of digital tools within MDT meetings are still rarely explored and demonstrated through prospective clinical trials. Initially, health information technology-based solutions were proposed to improve workflow and patient data management for MDTs, typically focusing on specific aspects of the process or targeting particular areas of application [8,10]. It is difficult to unify workflow models due to significant variation among the settings and protocols used for MDT meetings. Therefore, technical solutions (e.g., leveraging document templates) and process interoperability (e.g., toolkits for end-to-end collaboration and data sharing) may be helpful [11]. Recent digital tools for MDTs that address these issues such as the navify Tumor Board make it easy to access, collect, organize, and present information in a single dashboard, and have been reported to improve efficiency of MDTs and patient care [8,12,13,14,15,16,17].

The navify Tumor Board, launched in 2019, is a digital tool that streamlines MDT workflows by integrating patient data for efficient decision-making [12]. It consolidates imaging, pathology reports, and clinical notes into a single platform, reducing preparation and discussion time [18]. Despite offering significant efficiency gains, the tool involves costs for software licensing, staff training, and system integration, which must be balanced against its potential benefits in improving productivity and resource use in cancer care.

In Taiwan, the implementation of digital technologies within MDTs is not widespread, likely due to the challenges in accessing such solutions in the local market. Although some institutions have developed their own MDT systems, these are primarily used for meeting scheduling rather than serving as a platform for the structured gathering of medical data. This study aimed to evaluate whether the navify Tumor Board improves the efficiency of MDTs in the preparation, discussion, and follow-up stages compared to the conventional method.

## 2. Methods

### 2.1. Study Design and IRB

A prospective observational study was conducted at New Taipei Municipal TuCheng Hospital from January 2021 to June 2021. The time taken for preparation, discussion, and post-discussion follow-up in MDTs using the navify Tumor Board (navify group) was compared to that using the conventional method (conventional group). This study was approved by the Institutional Review Board of the Chang Gung Medical Foundation (No. 202100423B0).

### 2.2. Conventional Group

The “conventional method” refers to a manual, labor-intensive process that relies entirely on human effort. This approach involves independently recording patient identifiers, retrieving diagnostic data, reviewing all pathology findings (including those from metastatic sites), analyzing all imaging studies to identify a few key diagnostic images, manually determining cancer staging after discussion, confirming molecular test results, and searching for clinical trial information. Additionally, conclusions must be documented manually, records must be securely archived, and the responsible physicians must be notified. Follow-up actions, when necessary, are conducted during subsequent meetings. This workflow, characterized by its reliance on manual execution, demands substantial time, human resources, and cognitive capacity, highlighting its inefficiencies in modern clinical practice.

### 2.3. Stratification

Patients with newly diagnosed cancer, awaiting a treatment decision from MDTs, aged ≥18 years, and consenting to follow-up and chart reviews were included. MDT members included medical oncologists, radiologists, radiation therapy oncologists, nuclear medicine physicians, social workers, and nurse case managers. MDT meetings were performed every two weeks, with simple stratification (2:1) by cancer types, alternating sessions between one week using the navify Tumor Board and one week using the conventional method.

### 2.4. Phases of MDTs

In order to analyze the differences between the navify-based MDT group and the conventional group, the MDT process used in this study included four phases to better record the relevant details (Figure 1):(1)Phase A: Preparation of patient profiles with basic data and images for staging and sending meeting materials to MDT members (steps A-1 to A-4).(2)Phase B: Discussion including physician case presentation, images accessed and reviewed by radiologists, pathology and additional immunohistochemistry results reviewed by pathologists, expert comments, and conclusions regarding clinical decision-making (steps B-1 to B-6).(3)Phase C: Recording MDT comments in the EMRs (step C-1).(4)Phase D: Follow-up discussion schedule (step D-1). A follow-up MDT is required if there are any changes in the staging or treatment plan.

During the MDT process, two nurse case managers performed steps A-1 to A-4 and B-1. Two physicians performed steps B-2 to B-4 and B-6. The nurse case managers and physicians performed steps B-5, C-1, and D-1 (Appendix A). An independent observer used a standardized timer to record the time spent on each step according to the time and motion methodology for tracking the duration of various tasks [19]. The MDT determines the timing of the second discussion (phase D-1) during each meeting. The MDT evaluates whether a second discussion is necessary for each patient, particularly in complex cases where a specific targeted therapy is recommended and the team considers follow-up information to be critical. The second discussion is scheduled immediately following the MDT meeting when deemed necessary. This follow-up discussion typically occurs approximately three months after the completion of a standard course of anti-cancer therapy. The last follow-up date was 31 July 2024, at which point we counted the number of second discussions (or follow-up discussions) after the enrollment of each subject.

### 2.5. Software

The navify Tumor Board (Roche Molecular Systems, Belmont, CA, USA) is a cloud-based single platform that facilitates the arrangement of MDT meetings and the preparation, presentation, and recording of results for MDTs. The platform provides aggregated data such as diagnosis, cancer type, stage, biomarkers, key pathology, and radiology images, reducing manual data integration efforts instead of having clinicians hunt for relevant clinical data across disparate hospital data systems [1]. Additionally, the full integration of Clinical Decision Support apps within the navify Tumor Board can enhance multidisciplinary team discussions and support evidence-based treatment planning [20]. The navify Tumor Board was not integrated with our hospital’s data systems during the study period. Therefore, during phase A, the practitioners manually loaded patient profiles, images, histopathology, molecular diagnostics, and biomarkers into the navify Tumor Board. Every participating nurse case manager and physician received navify Tumor Board training before the study began.

### 2.6. Statistical Analysis

Descriptive statistics were used for patient characteristics expressed as a percentage and mean ± standard deviation (SD). Wilcoxon rank sum tests were used to determine the differences between groups, e.g., cancer stages. Time spent in all phases of MDTs using the navify Tumor Board and the conventional method was recorded in minutes and presented as mean ± SD and median (inter-quartile range [IQR]). The Wilcoxon rank sum test was performed to examine differences between the navify and conventional groups. *p*-values less than 0.05 were considered statistically significant. SAS software version 9.1.3 (SAS Institute Inc., Cary, NC, USA) was used for all analyses.

## 3. Results

A total of 377 new cancer cases were sent to the MDTs, with 237 in the navify group and 140 in the conventional group (Figure 2). Table 1 shows the basic characteristics of the enrolled patients. No significant differences were found in age, sex, cancer type, and stage between groups.

Of all the steps in phase A, checking all images to complete cancer staging (A-3) took the most time (mean = 2.39 ± 0.55 min; median = 2.13 [1.92–3.02] min). Compared to the conventional group, the navify group spent a longer time on the establishment of basic data (A-1) and collection of all images for cancer staging (A-2) but a shorter time on checking all images to complete cancer staging (A-3) and sending meeting materials to MDT members (A-4) (all *p*-values < 0.001). The overall preparation time was significantly reduced by 1.41% in the navify group compared to the conventional group (mean, 3.50 ± 0.31 vs. 3.83 ± 0.16 min; median, 3.55 [3.22–3.73] vs. 3.82 [3.70–3.92] min; *p* < 0.001) (Table 2).

Of all the steps in phase B, reviewing key images for discussion (B-3) took the longest time (mean = 5.09 ± 2.80 min; median = 4.38 [3.80–5.23] min). Compared to the conventional group, the navify group spent less time presenting patient history (B-2), reviewing pathology and additional immunohistochemistry results (B-4), and gathering comments from all relevant experts (B-5), but spent longer time concluding (B-6) (all *p*-values < 0.001). The time spent obtaining imaging access authority of EMRs (B-1; *p* = 0.701) and reviewing key images for discussion (B-3; *p* = 0.505) was comparable between both groups. The overall discussion time was significantly reduced by 21.26% in the navify group compared to the conventional group (mean, 8.00 ± 2.19 vs. 10.16 ± 4.45 min; median, 7.70 [6.18–9.35] vs. 8.98 [7.73–10.66] min; *p* < 0.001) (Table 2).

The navify group spent less time recording MDT comments (phase C) compared to the conventional group (mean, 2.02 ± 0.95 vs. 8.61 ± 2.60 min; median, 1.68 [1.25–2.97] vs. 8.35 [6.67–9.89] min; *p* < 0.001), achieving a time reduction of 76.54%. In phase D, only 69 cases in the navify group and 38 cases in the conventional group entered the second discussion. The navify group saved approximately one-third (33.43%) of the time required to schedule the second discussion (phase D) compared to the conventional group (mean, 4.74 ± 5.17 vs. 7.12 ± 7.24 min; median, 2.18 [1.52–5.85] vs. 3.73 [2.38–15.83] min; *p* = 0.011).

The overall time (the sum of all phases) spent on MDTs was significantly reduced by 35.37% in the navify group compared to the conventional group (mean, 18.40 ± 7.03 vs. 28.47 ± 10.78 min; median, 15.52 [12.88–22.97] vs. 25.11 [20.78–32.75] min; *p* < 0.001) (Table 2; Figure 3).

## 4. Discussion

The current study found that implementing the navify Tumor Board increases the efficiency of MDTs by reducing the time spent on meeting preparation, case discussion, recording comments, and scheduling the second discussion compared to the conventional method. Several studies have reported that the implementation of the navify Tumor Board for MDTs reduced case preparation and discussion time [8,12,13]. An observational study assessed case preparation time for breast cancer MDTs using the navify Tumor Board compared to the conventional method [8]. The results showed that a significant reduction in case preparation time using the navify Tumor Board was observed across specialties, including oncologists, radiologists, and surgeons (F = 71.74, *p* < 0.0001), and task categories (F = 38.98, *p* < 0.0001). Pathologists and pathology review tasks required comparable times using either method [8]. The authors indicated that retrieving pathology and radiology data from their information systems may involve steps, processes, and materials that a digital solution cannot directly impact [21]. Rather than expediting the review process, the navify Tumor Board may allow the radiologist or pathologist to focus more on reviewing the cases rather than on the data collection tasks. Another possible explanation was that the complexity of the pathology and radiology data was higher than that of the clinical course data (e.g., blood analytics) or other preparation tasks (e.g., checking meeting agendas), so a digital tool could not have a significant impact [8]. Another prospective study evaluated case preparation [12] and discussion [13] time pre- and post-navify Tumor Board implementation in MDTs for four types of cancer: breast, gastrointestinal (GI), head and neck, and hematopathology. The results showed that the navify Tumor Board significantly reduced case preparation time by 28%, 23%, and 33% in breast cancer (*p* = 0.036), GI (*p* = 0.041), and head and neck (*p* = 0.009) MDTs, respectively [12]. Case discussion time was significantly reduced by 27% and 20% in breast cancer (*p* < 0.05) and GI (*p* < 0.05) MDTs, respectively, using the navify Tumor Board [13]. There was no significant difference in the results for hematopathology, possibly because it was used to create and was the initial adopter of the navify Tumor Board, which resulted in variable results [12,13]. Although the case discussion time was not significantly improved in head and neck MDTs, case postponement rates were improved from 23% (pre-navify) to 10% (post-navify), possibly due to increased transparency in scheduling. Since postponement rates for other MDTs were already low (less than 5%), there was limited room for improvement [13]. In addition to time-saving, a reduction in the variance in case preparation and discussion time was also observed, demonstrating the benefits of standardizing meeting preparation and conduct processes using the navify Tumor Board [12,13].

Our findings are similar to those of previous studies, showing that implementing the navify Tumor Board saved time in the preparation and discussion phases. The navify Tumor Board has not been integrated into the preparation phase alongside our hospital data systems. Therefore, basic data and all images were manually loaded into the navify platform, resulting in more time spent in these phases than the conventional method. This is not surprising as adapting to new tools usually takes time initially due to the need for familiarity. However, this tool provides a standardization of structured data, which could lead to further time being saved in the long run. In contrast, the conventional method requires manually copying and pasting data, which lacks standardization. For case discussion, the navify Tumor Board reduced the overall case discussion time. It facilitated the presentation of case histories (step B-2), the review of pathological and immunohistochemistry results (step B-4), and the collection of comments from the relevant experts (step B-5). However, the improvement in the time taken to obtain the authority of EMRs (step B-1) and to review images for staging during MDT meetings (step B-3) was limited using the navify Tumor Board. It was likely that login steps to obtain the authority of EMRs could not be skipped due to information security reasons, and images for staging are required to be manually reviewed in our hospital. In the future, establishing an artificial intelligence model for image analysis in the navify Tumor Board may help image interpretation in MDTs. Our findings showed that implementing the navify Tumor Board also facilitates recording MDT comments in the EMRs (phase C) and scheduling the second discussion (phase D). Most of the cases entering the second discussion are complex. A Singaporean study of a complex case with dermatofibrosarcoma protuberans tumor found that implementing the navify Tumor Board reduced the number of steps required to prepare for and conduct MDT meetings by 46% and 31%, respectively. For the subspecialties in the MDTs (medical oncologists, radiation oncologists, surgeons, pathologists, etc.), the number of preparation steps performed by data managers and pathologists was reduced by 54% and 50%, respectively [14].

Besides the navify Tumor Board, several digital solutions for MDTs have been reported to facilitate the MDT process, including a reduction in preparation time and the number of cases (where diagnostic data are unavailable) prematurely sent for MDT discussion, an improvement in case postponement rates, and an increase in the number of healthcare professionals attending each meeting and taking on caseloads [15,16,17,22]. Digital tools were developed based on the needs of tumor boards, which vary across institutions due to a lack of standardization. Despite the lack of a standardized process model, several functional requirements for such digital tools to assist in MDTs were identified, including data privacy, automatic importing of case data, overviews of important clinical images related to cancer diagnosis and staging, annotations, the identification of similar historic cases, collaboration between different specialties, case presentation, and documentation [18]. The digital tools, including the navify Tumor Board, used in the studies mentioned above can provide a single platform that can meet these requirements, especially the automatic importing of case data, collaboration between clinicians and researchers, case presentation, and documentation, to become a one-stop-shop to facilitate MDT processes.

One critical point to address is that the final sample sizes do not align with the 2:1 ratio initially planned in the study design. This discrepancy arose from the application of simple stratification and the minor variations in case numbers that accumulated across groups. Following group allocation, some patients were excluded from the analysis for various reasons, including the absence of essential pathology reports (e.g., immunohistochemistry results), incomplete genetic testing, unfinished imaging procedures, or transfer to other healthcare facilities for treatment. These are common but unavoidable clinical factors that impacted the final dataset. Consequently, the cases included in the analysis, spanning phases A through D, deviated slightly from the initial distribution plan. For instance, the final analysis revealed ratios of 27:14 for breast cancer, 22:12 for colorectal cancer, and 59:27 for upper gastrointestinal cancers. To ensure the validity of the findings, a statistical analysis was conducted to confirm that the tumor group distributions did not exhibit significant differences (*p* = 0.616; Table 1), despite the use of simple stratification.

This prospective observational study offered solid evidence of how a digital tool can significantly improve the efficiency of MDTs in Taiwan. A large sample of 377 cases, with 237 using the navify Tumor Board and 140 using the conventional method, provided a robust comparison of the two approaches. The detailed evaluation of tasks and activities in each phase allowed for a clear understanding of how the navify Tumor Board facilitated the MDT process and standardized tumor board preparation, data retrieval, presentation, documentation, and follow-up. The limitations of this study include the fact that the study period was six months. Therefore, the number of cases that entered the second discussion phase was small. Second, this was a single-center study, so our findings cannot be generalized to all institutions that used different MDT processes/protocols or digital tools. Third, this study demonstrated a statistically significant difference in time between the two groups, with the navify group showing a reduction in time, allowing physicians to focus more on discussing patient conditions and formulating personalized strategies. However, the study did not include clinical endpoints such as patient satisfaction or physician satisfaction. We hope that future research will incorporate such survey components to enhance the comprehensiveness of the findings. Third, the navify Tumor Board has not been fully integrated into the hospital information systems in this study, highlighting the need for further investigation. Unlike in-house-developed digital tools, the tools used in this study require external funding beyond the scope of clinical trials. Therefore, exploring how this tool can assist clinicians and nurse case managers in all MDT meetings requires a longer observation period and broader application across more multidisciplinary teams, rather than being limited to the context of clinical trials. Interoperability with various hospital information systems will be essential in the future. We hope that implementing similar digital solutions will soon lead to tangible clinical benefits and increased efficiency for hospitals, oncologists in MDTs, and cancer patients.

## 5. Conclusions

This large-sample prospective observational study provided evidence that the digital tool navify Tumor Board can significantly improve the efficiency of MDT workflows in Taiwan.

## Figures and Tables

**Figure 1 cancers-17-00444-f001:**
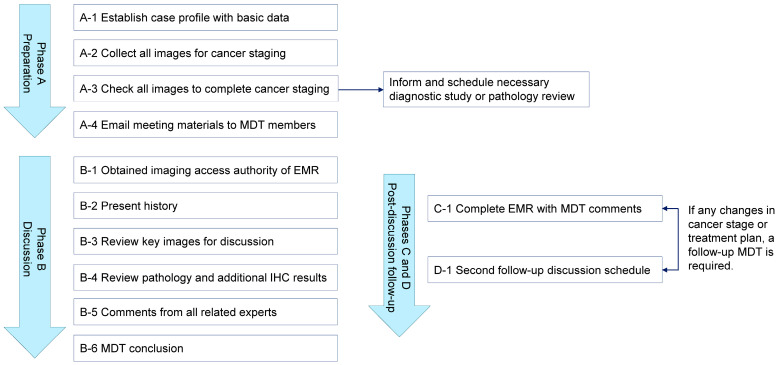
Diagram of the MDT process. EMR, electronic medical record; IHC, immunohistochemistry; MDT, multidisciplinary tumor board.

**Figure 2 cancers-17-00444-f002:**
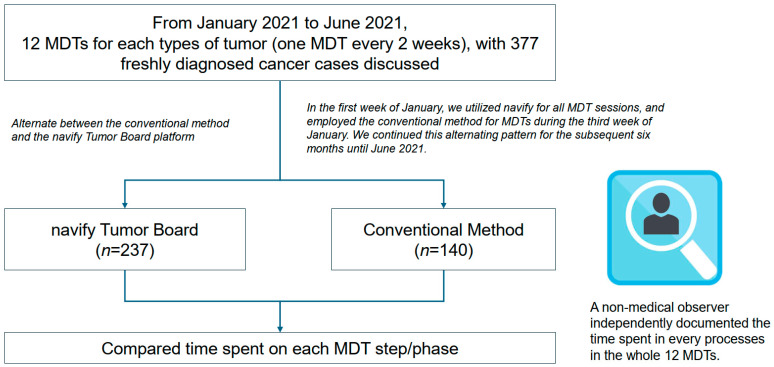
Study flow diagram. MDT, multidisciplinary tumor board.

**Figure 3 cancers-17-00444-f003:**
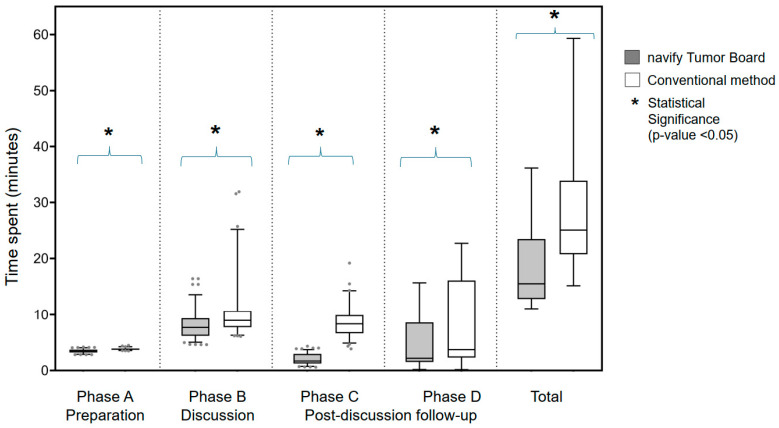
Box plots for time spent on all phases for MDTs using the navify Tumor Board or the conventional method. Values outside the range between 2.5 and 97.5 percentiles are shown as dots. For phase D and total phases, 69 cases (29.11%) and 38 (27.14%) cases entering the second discussion in the navify and conventional groups, respectively, were included in the analysis. MDT, multidisciplinary tumor board.

**Table 1 cancers-17-00444-t001:** Basic characteristics.

	Navify Tumor Board (*n* = 237)		Conventional Method (*n* = 140)		
	*n*	%	*n*	%	*p*-Value ^a^
Age, years (mean ± SD)	57.30 ±18.10		58.10 ± 19.10		0.130
Sex					
Male	136	57.40%	76	54.30%	
Female	101	42.60%	64	45.70%	0.558
Cancer types					
Breast cancer	27	11.40%	14	10.00%	
Colorectal cancer	22	9.30%	12	8.60%	
Upper gastrointestinal cancers, including liver, pancreas	59	24.90%	27	19.30%	
Genitourinary and gynecologic cancer	17	7.20%	10	7.10%	
Head and neck cancer, including thyroid	20	8.40%	9	6.40%	
Others, including hematologic cancer and soft tissue sarcomas	10	4.20%	5	3.60%	
Lung cancer	82	34.60%	63	45.00%	0.616
Staging					
Early stage	99	41.80%	62	44.30%	
Advanced stage	138	58.20%	78	55.70%	0.634

^a^ Wilcoxon rank sum tests were performed to examine differences between two groups. SD, standard deviation.

**Table 2 cancers-17-00444-t002:** Descriptive statistics for time spent in all phases and steps.

	Total (*n* = 377)		Navify Tumor Board (*n* = 237)		Conventional Method (*n* = 140)		
Phases/Steps (min)	Mean (SD)	Median (IQR)	Mean (SD)	Median (IQR)	Mean (SD)	Median (IQR)	*p*-Value ^a^
Preparation Phase A							
A-1 Establish case profile with basic data	0.33 (0.13)	0.32 (0.20–0.42)	0.41 (0.10)	0.40 (0.32–0.45)	0.20 (0.03)	0.20 (0.18–0.22)	<0.001
A-2 Collect all images for cancer staging	0.65 (0.31)	0.72 (0.30–0.88)	0.86 (0.17)	0.82 (0.77–0.95)	0.28 (0.03)	0.28 (0.27–0.30)	<0.001
A-3 Check all images to complete cancer staging	2.39 (0.55)	2.13 (1.92–3.02)	1.98 (0.20)	1.98 (1.85–2.08)	3.07 (0.14)	3.07 (2.95–3.13)	<0.001
A-4 Email meeting materials to MDT members	0.26 (0.03)	0.25 (0.23–0.28)	0.24 (0.03)	0.23 (0.22–0.25)	0.28 (0.02)	0.28 (0.27–0.30)	<0.001
Subtotal	3.62 (0.31)	3.68 (3.42–3.83)	3.50 (0.31)	3.55 (3.22–3.73)	3.83 (0.16)	3.82 (3.70–3.92)	<0.001
Discussion Phase B							
B-1 Obtained imaging access authority of EMR	0.28 (0.16)	0.23 (0.22–0.28)	0.31 (0.20)	0.23 (0.20–0.33)	0.24 (0.03)	0.23 (0.22–0.25)	0.701
B-2 Present history	0.76 (0.28)	0.77 (0.53–0.97)	0.62 (0.21)	0.58 (0.47–0.77)	1.01 (0.21)	1.02 (0.86–1.13)	<0.001
B-3 Review key images for discussion	5.09 (2.80)	4.38 (3.80–5.23)	4.87 (1.71)	4.47 (3.65–5.57)	5.45 (4.01)	4.33 (3.98–4.82)	0.505
B-4 Review pathology and additional IHC results	0.93 (0.78)	0.60 (0.45–1.12)	0.63 (0.37)	0.50 (0.42–0.65)	1.44 (0.99)	1.13 (0.63–2.08)	<0.001
B-5 Comments from all related experts	1.00 (0.75)	0.80 (0.53–1.13)	0.74 (0.36)	0.77 (0.47–0.92)	1.44 (0.99)	1.13 (0.63–2.08)	<0.001
B-6 MDT conclusion	0.74 (0.39)	0.70 (0.43–0.93)	0.84 (0.35)	0.87 (0.63–0.98)	0.58 (0.41)	0.53 (0.36–0.63)	<0.001
Subtotal	8.80 (3.38)	8.17 (6.80–9.97)	8.00 (2.19)	7.70 (6.18–9.35)	10.16 (4.45)	8.98 (7.73–10.66)	<0.001
Post-discussion follow-up Phase C							
C-1 Complete EMR with MDT comments	4.47 (3.64)	3.15 (1.38–7.07)	2.02 (0.95)	1.68 (1.25–2.97)	8.61 (2.60)	8.35 (6.67–9.89)	<0.001
Phase D							
D-1 Second follow-up discussion schedule ^b^	5.59 (6.06)	2.90 (1.52–11.35)	4.74 (5.17)	2.18 (1.52–5.85)	7.12 (7.24)	3.73 (2.38–15.83)	0.011
Total ^b^	21.97 (9.78)	19.32 (14.60–27.78)	18.40 (7.03)	15.52 (12.88–22.97)	28.47 (10.78)	25.11 (20.78–32.75)	<0.001

^a^ Wilcoxon rank sum test was performed to examine differences between two groups. ^b^ A total of 69 cases and 38 cases entered the second discussion in the navify and conventional groups, respectively. Analyses were performed with available cases. EMR, electronic medical record; IHC, immunohistochemistry; IQR, inter-quartile range; MDT, multidisciplinary tumor board; SD, standard deviation.

## Data Availability

The study data can be requested upon reasonable request, but access is subject to approval by the IRB.

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
