# Peer review of "The Use of an Integrated Digital Tool to Improve the Efficiency of Multidisciplinary Tumor Boards—A Prospective Trial in Taiwan"

_cancers, 2025, doi:10.3390/cancers17030444_

Round 1

Reviewer 1 Report

Comments and Suggestions for Authors

Review for cancers-3392192

In this original research article entitled “Use of An Integrated Digital Tool to Improve the Efficiency of Multidisciplinary Tumor Boards – A Prospective Trial in Taiwan”, the authors (Chang et al.) studied the impact of the Tumor Board digital tool on multidisciplinary team tumor boards (MDT) efficiency and compared its streamlined workflow and preparation to conventional, time-consuming methods. The authors reported that the navify Tumor Board can significantly improve the efficiency of MDT workflow in Taiwan.

Hereafter some revealed comments after reviewing the manuscript.

1-       The major lack of the study is the single-center assessment, which makes the findings not valid for all Taiwan hospitals and/or different countries.

2-       The extrapolation of the digital findings into possible clinical outcomes should be discussed and not only reported in the conclusion.

3-       The sentence “The authors indicated … solution cannot directly impact.” can be supported by the following relevant and recent publication; doi: 10.1007/s00223-021-00931-3

4-       It is recommended to add asterisk(s) or letter(s) to the box plots to exhibit any significant statistical differences, specifically in figure 3.

5-       English language is overall acceptable but there are several spelling mistakes just like “aavify” in line 21.

Author Response

Reviewer #1

Comment #1-1: The major lack of the study is the single-center assessment, which makes the findings not valid for all Taiwan hospitals and/or different countries.

Response #1-1

We completely agree with your comment regarding the lack of validation of our study in other hospitals in Taiwan. We have listed it as the second limitation of ths present study. (Page 10, line 302-307) The primary reason for conducting this study was to evaluate the Navify Tumor Board. This mature commercial digital tool has not yet been tested through a prospective trial, even in a single medical center. While our current findings are limited to a single hospital, we firmly believe they should be replicated and externally validated by other institutions.

In Taiwan, we are planning another prospective trial involving the Navify Tumor Board across seven branches of the Chang Gung Medical System located in northern and southern Taiwan. This broader study will likely provide a more representative sample and facilitate the generalizability of the findings across multiple medical centers in Taiwan.

Despite the limitation of being a single-center study, our research includes two significant highlights: (1) Comprehensive Analysis of Tumor Board Operations - For the first time, to the best of our knowledge, we systematically delineated the operation of a multidisciplinary tumor board into 4 phases (A to D). Through this detailed analysis, we were able to identify the specific stages where a digital tool like the Navify Tumor Board provides the most benefit. This approach offers a novel and systematic evaluation of digital tools for improving tumor board operations. (2) Prospective Study Design - Unlike previous studies, which were mostly limited to small case series or observational designs, our research employs a prospective study design. We were pleasantly surprised to discover that a well-designed digital tool can significantly enhance the efficiency of multidisciplinary tumor boards, reducing manpower and resource usage.

We believe that reporting these findings early to the scientific community will provide a framework for future studies on similar digital tools. Our systematic approach can serve as a reference for analyzing the impact of such tools on team operations and for demonstrating their effectiveness in improving efficiency.

Comment #1-2: The extrapolation of the digital findings into possible clinical outcomes should be discussed and not only reported in the conclusion.

Response #1-2

We fully agree with your insightful comment. We have revised both the discussion and conclusions to address this concern more thoroughly. The revised sections now read as follows:

"….In this study, the navify Tumor Board has not been fully integrated with the hospital information systems, which warrants further studies. Interoperability with different hospital information systems is needed in the future. Interoperability with various hospital information systems will be essential in the future. We hope implementing similar digital solutions will soon lead to tangible clinical benefits and increased efficiency for hospitals, oncologists in multidisciplinary teams (MDTs), and cancer patients.

  1. Conclusions

This large-sample prospective observational study provided evidence that the digital tool navify Tumor Board can significantly improve the efficiency of MDT workflow in Taiwan. We hope that implementing similar digital solutions will soon translate into direct clinical benefits and efficiency for hospitals, oncologists in the MDT, and cancer patients."

These changes are reflected in lines 309-320 on page 10 of the revised manuscript. This expanded discussion could now better address the extrapolation of our findings to potential clinical outcomes. 

Comment #1-3: The sentence "The authors indicated … solution cannot directly impact." can be supported by the following relevant and recent publication; doi: 10.1007/s00223-021-00931-3

Response #1-3

Thank you for the suggestion. We have added the referenced publication to support the sentence and strengthen our work.

The revised sentence is now:

"The authors indicated that pathology and radiology data retrieval from their information systems may involve steps, processes, and materials that a digital solution cannot directly impact.[20]

  1. Badraoui, R.; Saeed, M.; Bouali, N.; Hamadou, W.S.; Elkahoui, S.; Alam, M.J.; Siddiqui, A.J.; Adnan, M.; Saoudi, M.; Rebai, T. Expression profiling of selected immune genes and trabecular microarchitecture in breast cancer skeletal metastases model: Effect of α–tocopherol acetate supplementation. Calcified Tissue International 2022, 1-14.

Comment #1-4: It is recommended to add asterisk(s) or letter(s) to the box plots to exhibit any significant statistical differences, specifically in figure 3.

Response #1-4

Thank you for the suggestion. We have revised the figures by adding asterisk(s) to clearly illustrate the statistical significance, with particular attention to Figure 3, as follows:

Comment #1-5: English language is overall acceptable but there are several spelling mistakes just like "aavify" in line 21.

Response #1-5

Thank you for your careful review. We have thoroughly revised and carefully examined the entire manuscript for spelling errors. The misspelling "aavify" has been corrected to "navify."

The revised sentence now reads:

"PURPOSE This study assessed the impact of the navify Tumor Board digital tool on multidisciplinary team tumor boards (MDT) efficiency, comparing its streamlined workflow and preparation to conventional, time-consuming methods. "

We appreciate your valuable feedback in helping us improve the manuscript.

Reviewer 2 Report

Comments and Suggestions for Authors

The article describes a formal test of a digital system called "navify" to facilitate the work of molecular tumor

boards(MTBs). The authors concluded based on rigorous analysis that navify speeds up the MTB work compared to preceding methods.

The study is well-designed and the technical analysis is done well. However, some of the writing could be improved.

Major comments:

1. The Introduction needs to have a few additional paragraphs early on explaining what navify does, explaining the costs, and citing much earlier what is now reference 19.

2. I do not understand what "simple stratification (2:1) by cancer type"  means in this study,

The cases were assigned to the two arms at a ratio of 237:140, which is closer to a proportion of 3:2 and 2:1.

3. The text about second discussions of a patient near line 101 and describing part D of Table 2 needs to be substantially expanded to answer questions such as:

Was it decided at the end of each meeting which patients who had a first discussion at a meeting would require

a second discussion? If not, how was the need for a second discussion determined?

Did any patient require a third discussion?

Since the MTB alternated between navify and conventional meetings, does this imply that a patient who needed a second discussion would have to wait two weeks for the second discussion? If so, did the delay ever cause clinical problems?

How did the authors account for patients whose first discussion took place in the last two weeks of the trial and needed a second discussion?

My understanding is that the second discussion would take place after the formal end of the study in June 2021, so it is not clear whether the time for the second discussion is counted.

4. I did not see a clear statement indicating whether after the study, the Nw Taipei Municipal TuCheng Hospital adopted navify for all subsequent MTB sessions. If not, the reasons for not switching fully to navify need to be explained.

Comments on the Quality of English Language

The manuscript has some errors in word choice and grammar that should be corrected after the above major points are addressed.

line 21, change "aavify" to "navify" [typo]

line 35, change "navify Tumor Board" to "navify Tumor Board tool" [grammar]

line 41, change "Multidisciplinary oncology care team held" to "Multidiciplinary care teams hold" [grammar]

line 46, change "cumbrous" to "cumberosome" [wrong word]

line 57, change "MDT" to "MDTs" [grammar]

line 80, change "was compared" to "were compared"

line 104-105, change "navify-based MDT and conventional group" to "the navify-based MDT group and the conventional group" [grmmar]

line 108, change "Discussion includes" to "Discussion including"

line 120, change "Besides, an independent observer" to "An independent observer"

line 139, change "i.e., cancer stages" to "e.g., cancer stages" [wrong Latin abbreviation]

line 141, change "test was" to "tests were"

line 206, "comparable time" to "comparable times" [grammar]

line 228, change "conduction" to "conduct" [wrong word]

Line 282, change "NAVIFY" to "navify" [for consistency with other uses]

Lines 335-336, reference 17 is incomplete, lacking either page numebrs or a manuscript number

Author Response

Reviewer #2

The article describes a formal test of a digital system called "navify" to facilitate the work of molecular tumor boards(MTBs). The authors concluded based on rigorous analysis that navify speeds up the MTB work compared to preceding methods. The study is well-designed and the technical analysis is done well. However, some of the writing could be improved.

Major comments:

Comments #2-1. The Introduction needs to have a few additional paragraphs early on explaining what navify does, explaining the costs, and citing much earlier what is now reference 19.

Response #2-1

We thank the reviewer for the helpful comments. We have added a paragraph in the Introduction to provide a brief description of the navify Tumor Board, along with information on its development and costs, and have cited the relevant reference earlier. The revised sentences now read as follows:

"The Navify Tumor Board, launched in 2019, is a digital tool that streamlines mul-tidisciplinary tumor board (MDT) workflows by integrating patient data for efficient decision-making.[18] It consolidates imaging, pathology reports, and clinical notes into a single platform, reducing preparation and discussion time.[19] While offering signif-icant efficiency gains, the tool involves costs for software licensing, staff training, and system integration, which must be balanced against its potential benefits in improving productivity and resource use in cancer care."

These sentences have been added to the Introduction on lines 82-87 (page 2), providing essential background before the study details.

Comments #2-2.

I do not understand what "simple stratification (2:1) by cancer type"  means in this study. The cases were assigned to the two arms at a ratio of 237:140, which is closer to a proportion of 3:2 and 2:1.

Responses #2-2

Thank you for your valuable suggestions. The final analysis numbers do not appear to be in a 2:1 ratio, but this is the result of simple stratification and the small differences in case numbers that accumulated across the groups. Some patients, after being assigned to groups, were unable to complete the discussion for various reasons (e.g., missing pathology reports such as immunohistochemistry, incomplete genetic testing, unfinished imaging, or the patient being transferred to another hospital for treatment, among other common but uncontrollable clinical factors). As a result, the final cases included in the analysis, covering phases A to D, deviated slightly from the original setting. For example, the final analysis for breast cancer (27:14), colorectal cancer (22:12), and upper gastrointestinal cancers (59:27) reflected similar issues. This is why, despite having simple stratification, we still needed an analysis to confirm that the distribution of tumor groups showed no significant differences (p=0.616, Table 1).

Comments #2-3.

The text about second discussions of a patient near line 101 and describing part D of Table 2 needs to be substantially expanded to answer questions such as:

  • Was it decided at the end of each meeting which patients who had a first discussion at a meeting would require a second discussion? If not, how was the need for a second discussion determined? Did any patient require a third discussion?
  • Since the MTB alternated between navify and conventional meetings, does this imply that a patient who needed a second discussion would have to wait two weeks for the second discussion? If so, did the delay ever cause clinical problems?
  • How did the authors account for patients whose first discussion took place in the last two weeks of the trial and needed a second discussion?

My understanding is that the second discussion would take place after the formal end of the study in June 2021, so it is not clear whether the time for the second discussion is counted.

Response #2-3

  • Yes, as you realized, the MDT members would decide whether a second discussion is necessary for each patient in each MDT meeting. For example, in the case of a patient with clinical stage II colorectal cancer discussed in MDT before curative surgery with imaging indeterminate lymph node involvement status (very common), the experts may request a second discussion after the surgery to determine if adjuvant cancer therapy is required. Another example could involve a stage 4 lung patient with a rare genetic mutation. Suppose the MDT recommends a specific targeted therapy, and the team deems the follow-up information crucial. In that case, a second round of discussion will be scheduled right after the MDT meeting. The second round of discussion takes place typically around three months later, after completing a standard course of anti-cancer therapy.
  • No, the treatment will be carried out based on the first discussion. A second discussion does not mean the patient should wait for treatment until after the second discussion. A second discussion is scheduled for additional therapies that may be required following the initial treatment. For example, in the case mentioned earlier, where the lymph node status is unclear before surgery and adjuvant therapy is deemed essential, a second discussion would be raised to decide on the need for chemotherapy after surgery. However, the surgery would not be delayed while waiting for the second discussion to be scheduled.
  • Certainly, this is an important question. Not every patient requires a second discussion, especially when the treatment recommendation is straightforward and simple. However, for special cases requiring additional recommendations, a second discussion (or follow-up discussion) is clinically useful for personalized medicine in complex cases. As mentioned in the study results, "In phase D, only 69 cases in the Navify group and 38 cases in the conventional group entered the second discussion." When the study concluded and enrollment ended, we counted the number of second discussion events for each enrolled patient until the last follow-up date of the study. To clarify the cutoff date for counting second discussions for enrolled cases, we have added previously missing information in the methods section. The revised text now reads: "The last follow-up date was 31st July 2024, at which point we counted the number of second discussions (or follow-up discussions) after the enrollment for each subject." (lines 142-143, Page 4).

Comments #2-4. I did not see a clear statement indicating whether after the study, the Nw Taipei Municipal TuCheng Hospital adopted navify for all subsequent MTB sessions. If not, the reasons for not switching fully to navify need to be explained.

Response #2-4

We thank you for the valuable and practical question. As this report is based on a prospective trial, we are pleased to present the outcomes demonstrating how digital tools can improve and facilitate MDT processes. However, unlike in-house developed digital tools, the tools we used in this study require external funding beyond the scope of clinical trials. Therefore, we are still exploring how this tool can assist clinicians and nurse case managers at New Taipei Tucheng Hospital, albeit in the context of many clinical trials.

Comments #2-5 Comments on the Quality of English Language

The manuscript has some errors in word choice and grammar that should be corrected after the above major points are addressed.

  • line 21, change "aavify" to "navify" [typo] (Corrected)
  • line 35, change "navify Tumor Board" to "navify Tumor Board tool" [grammar] (Corrected)
  • line 41, change "Multidisciplinary oncology care team held" to "Multidiciplinary care teams hold" [grammar] (Corrected)
  • line 46, change "cumbrous" to "cumberosome" [wrong word] (Corrected)
  • line 57, change "MDT" to "MDTs" [grammar] (Corrected)
  • line 80, change "was compared" to "were compared" (Corrected)
  • line 104-105, change "navify-based MDT and conventional group" to "the navify-based MDT group and the conventional group" [grmmar] (Corrected)
  • line 108, change "Discussion includes" to "Discussion including" (Corrected)
  • line 120, change "Besides, an independent observer" to "An independent observer" (Corrected)
  • line 139, change "i.e., cancer stages" to "e.g., cancer stages" [wrong Latin abbreviation] (Corrected)
  • line 141, change "test was" to "tests were" (Corrected)
  • line 206, "comparable time" to "comparable times" [grammar] (Corrected)
  • line 228, change "conduction" to "conduct" [wrong word] (Corrected)
  • Line 282, change "NAVIFY" to "navify" [for consistency with other uses] (Corrected)
  • Lines 335-336, reference 17 is incomplete, lacking either page numebrs or a manuscript number (Corrected)

Responses #2-5 

All the errors and typos have been corrected in the revised manuscript. We are very grateful to have such a careful reviewer. Thank you so much!

Round 2

Reviewer 2 Report

Comments and Suggestions for Authors

All the authors' changes are fine.

All the authors' responses are clear and thorough and satisfactory in their substance. 

However, the authors' responses 2-2, 2-3, 2-4 were not incorporated in any way to the manuscript. Three additions should be made to the manuscript to communicate to all readers the content of responses 2-2, 2-3, 2-4.

Author Response

Point-to-point Responses to Reviewers' Comments

Comments #1 All the authors' responses are clear and thorough and satisfactory in their substance. However, the authors' responses 2-2, 2-3, 2-4 were not incorporated in any way to the manuscript. Three additions should be made to the manuscript to communicate to all readers the content of responses 2-2, 2-3, 2-4.

Response #1

We apologize for not incorporating the responses into the revised manuscript. We have tried to do that in this version of the revised manuscript.

For previous responses #2-2, to address this point, we have added the following paragraph to the Discussion section:

"One critical point to address is that the final sample sizes do not align with the 2:1 ratio initially planned in the study design. This discrepancy arises from the application of simple stratification and the minor variations in case numbers that accumulated across groups. Following group allocation, some patients were excluded from the analysis for various reasons, including the absence of essential pathology reports (e.g., immunohistochemistry results), incomplete genetic testing, unfinished imaging procedures, or transfer to other healthcare facilities for treatment. These are common but unavoidable clinical factors that impacted the final dataset. Consequently, the cases included in the analysis, spanning phases A through D, deviated slightly from the initial distribution plan. For instance, the final analysis revealed ratios of 27:14 for breast cancer, 22:12 for colorectal cancer, and 59:27 for upper gastrointestinal cancers. To ensure the validity of the findings, a statistical analysis was conducted to confirm that the tumor group distributions did not exhibit significant differences (p = 0.616; Table 1), despite the use of simple stratification."

(Lines 300–313, Pages 10–11)

For previous responses #2-3, we have added the following paragraph to the Methods section:

"The MDT determines the timing of the second discussion (phase D-1) during each meeting. The MDT evaluates whether a second discussion is necessary for each patient, particularly in complex cases where a specific targeted therapy is recommended, and the team considers follow-up information critical. The second discussion is scheduled immediately following the MDT meeting when deemed necessary. This follow-up discussion typically occurs approximately three months after the completion of a standard course of anti-cancer therapy."  (Lines 140–147, Pages 4)

For previous responses #2-4, to address this limitation, we have added the following paragraph to the Discussion section:

"Third, the navify Tumor Board has not been fully integrated with the hospital information systems in this study, highlighting the need for further investigation. Unlike in-house-developed digital tools, the tools used in this study require external funding beyond the scope of clinical trials. Therefore, exploring how this tool can assist clinicians and nurse case managers in all MDT meetings requires a longer observation period and broader application across more MDT board meetings rather than being limited to the context of clinical trials. " (Lines 329–335, Page 10)
